# Collaborative Large Language Model for Recommender Systems

## ABSTRACT

Recently, there is a growing interest in developing next-generation recommender systems (RSs) based on pretrained large language models (LLMs), fully utilizing their encoded knowledge and reasoning ability. However, the semantic gap between natural language and recommendation tasks is still not well addressed, leading to multiple issues such as spuriously-correlated user/item descriptors, ineffective language modeling on user/item contents, and inefficient recommendations via auto-regression, etc. In this paper, we propose **CLLM4Rec**, the first generative RS that tightly integrates the LLM paradigm and ID paradigm of RS, aiming to address the above challenges simultaneously. We first extend the vocabulary of pretrained LLMs with user/item ID tokens to faithfully model the user/item collaborative and content semantics. Accordingly, in the pretraining stage, a novel *soft+hard prompting* strategy is proposed to effectively learn user/item collaborative/content token embeddings via language modeling on RS-specific corpora established from user-item interactions and user/item features, where each document is split into a prompt consisting of heterogeneous *soft* (user/item) tokens and *hard* (vocab) tokens and a main text consisting of homogeneous item tokens or vocab tokens that facilitates stable and effective language modeling. In addition, a novel mutual regularization strategy is introduced to encourage the CLLM4Rec to capture recommendation-oriented information from user/item contents. Finally, we propose a novel recommendation-oriented finetuning strategy for CLLM4Rec, where an item prediction head with multinomial likelihood is added to the pretrained CLLM4Rec backbone to predict hold-out items based on the soft+hard prompts established from masked user-item interaction history, where recommendations of multiple items can be generated efficiently. Experiments on multiple real-world datasets show the superior effectiveness and efficiency of CLLM4Rec compared with state-of-the-art[1].

## 1 INTRODUCTION

With content growing exponentially on the Web, recommender system (RS) has become an essential component for online service platforms [12]. Nevertheless, since Netflix released its Prize in 2006 [2], RS has long been dominated by the ID-based paradigm, where users and items are represented by unique, continuous ID embeddings denoting their semantic similarity (e.g., w.r.t. users' preferences on items, user/item contents, etc.) [32]. Exemplar ID-based RSs include matrix factorization-based methods such as PMF [23] and the two-tower models [30], where the user/item ID embeddings are either randomly initialized and learned from their historical interactions (i.e., collaborative filtering [15]), or established based on user/item content features (i.e., content-based methods [20]).

Recently, large language model (LLM) has become a heated research topic that revolutionized both academia and industry [34]. Transformer-based neural networks with billions of parameters [28], such as GPT [24], T5 [25], LlaMA [27], have demonstrated

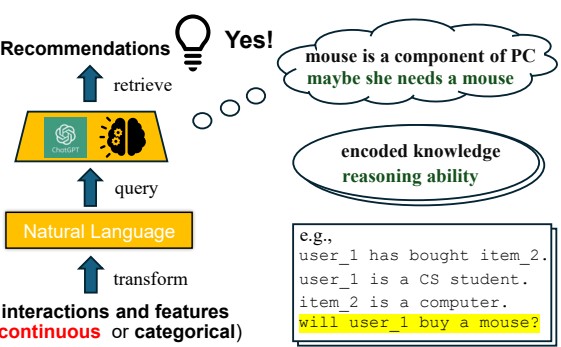

**Figure 1: Prospective of developing the next generation of recommender systems based on the pretrained LLMs.**

**emergent ability** when trained on large-scale corpora [29], showcasing an unprecedented understanding of knowledge and patterns contained in natural language [34]. Consequently, it is promising to develop the next generation of RS based on the pretrained LLMs [5], fully utilizing their encoded knowledge, logical reasoning ability, and generative AI power to understand and reason with the user/item semantics and make more accurate recommendations accordingly, especially when users and items are associated with large amounts of textual features, such as biographies, descriptions, content, reviews, and explanations, etc., in modern online platforms [21]. (see Fig. 1 for an intuitive example of an LLM-based RS)

Several preliminary studies have been conducted to investigate the adaptation of LLMs for recommendation systems [4, 7, 16]. Typically, these methods can be summarized into two steps: **1)** First, instead of representing users/items with continuous ID embeddings, relevant information necessary for reasoning with user interests and generating recommendations, i.e., target user, interacted items, user/item features, and candidate items, are converted into a *natural language-based prompt.* **2)** Then, the prompt is used to query the LLM, where information relevant to recommendations (e.g., whether the user will interact with an item or not) is retrieved from the *textual output* of the LLM to generate recommendations. The above procedure can be performed in a zero-shot manner [6, 8, 9, 33], where the recommendation decisions are obtained directly from the pretrained LLM (e.g., we input all relevant information regarding a user and an item into the chatbox of ChatGPT and ask if the user will interact with the item), or if groundtruths are available, the pretrained LLMs can also be finetuned, such that RS-specific knowledge can be updated into the pretrained model [3, 7, 13, 31].

Although progress has been achieved by these pioneer works, some fundamental dichotomies between natural language processing (NLP) and recommendation still remain to be addressed. One main challenge is the gap between natural language and user/item semantics. Generally, there are two strategies to represent user/item in an LLM-based RS. One strategy is the pseudo-ID-based method, where an ID-like word (e.g., "user_*i*" or "item_*j*") is used to represent the *i*th user and *j*th item [7]. However, since the vocabulary of most

---

[1]Codes are anonymously released in this URL.

LLM contains number-tokens up to two digits, when tokenized, the pseudo ID breaks down into atomic tokens, e.g., "user_4332" into ["user", "_", "43", "32"], where spurious correlations can be introduced for irrelevant users/items (e.g., "user_4332" with "user_43" and "user_32"). In contrast, description-based methods use semantically meaningful descriptions to index users/items, such as item titles [4, 9] or a small amount of newly-introduced tokens assigned to different user/items based on their content similarity [11]. However, description-based methods introduce a strong inductive bias on user-item semantic similarity, which may not faithfully capture the true semantics. Introducing user/item ID tokens, unfortunately, is generally considered infeasible for pretrained LLMs, as directly conducting language modeling on sequences with heterogeneous tokens can be ineffective and unstable, especially when the vocabulary of most LLMs is diluted (e.g., $\sim$ 50k for GPT, and $\sim$ 30k for T5) by a large number of randomly initialized user/item embeddings.

Even if user/item ID token embeddings can be effectively learned via language modeling, another challenge that hinders effective collaborative filtering with LLMs is that, since the order of interactions usually does not matter for direct recommendations while human language naturally has an order, spurious temporal correlation can be introduced for items placed in different positions when transforming the user historical interactions into textual sentences. Furthermore, for content modeling, since pretrained LLMs are not recommendation-oriented, they can easily capture noise in the user/item textual features irrelevant to the recommendation purpose. Finally, since LLMs generate the next token in an autoregressive manner, recommending multiple items can be inefficient. For both pseudo-ID-based and description-based indexing strategies, item candidates usually need to be explicitly provided in the prompt. These issues severely hinder their industrial applications where the candidate pool is large and low latency matters.

To address the above challenges, we present **CLLM4Rec**, the first method that tightly combines the ID paradigm of RS with the LLM-based paradigm to address the semantic gap. We first extend the vocabulary of pretrained LLMs with user/item ID tokens to faithfully model the user/item collaborative/content semantics, where the embeddings are learned in two stages. The *pretraining stage* consists of mutually-regularized collaborative and content LLMs that learn user/item token embeddings via language modeling on RS-specific corpora established from user/item interactions and textual features. Specifically, a novel "soft+hard" prompting strategy is proposed for effective language modeling on documents with heterogeneous tokens, where each document is decomposed into a prompt consisting of user/item (*soft*) and vocab (*hard*) tokens that describe the contexts and a main text consisting of homogeneous item tokens (i.e., interaction history) or vocab tokens (i.e., user/item textual features), respectively. Through this strategy, the prediction heads for the two LLMs can focus exclusively on collaborative and content information, and the stability and effectiveness of language modeling can be substantially enhanced. In addition, a stochastic reordering strategy is proposed for the collaborative LLM to ignore the order of item tokens without negative influence on the vocab tokens. Finally, we propose a novel recommendation-oriented *finetuning strategy* for CLLM4Rec, where an item prediction head with multinomial likelihood is added to the pretrained collaborative LLM backbone to predict hold-out items based on soft+hard

prompts established from masked users' interaction history, where recommendations of multiple items can be generated efficiently. The contribution of this paper can be concretely summarized as:

- We present CLLM4Rec, the first framework that tightly couples the ID paradigm and LLM paradigm of RS, where encoded knowledge and reasoning ability of LLMs can be fully utilized, while user/item ID token embeddings aligned to the vocab space can well capture intrinsic user interests and item properties.
- A novel soft+hard prompting strategy is proposed to pretrain the LLMs on sequences of heterogeneous tokens describing user historical interactions and user/item features via language modeling, where the collaborative and content information can be effectively learned by the user/item token embeddings.
- A mutual-regularization strategy is proposed to constrain the CLLM4Rec to learn information more relevant for recommendations from user/item content. In addition, stochastic reordering is proposed such that the order of item tokens can be ignored by the collaborative LLM without influence on the textual parts.
- A recommendation-oriented finetuning strategy is proposed for CLLM4Rec, where an item prediction head with multinomial likelihood is added on the collaborative LLM that predicts hold-out items based on prompt interaction history, where recommendations for multiple items can be generated efficiently.

## 2 RELATED WORK

### 2.1 Large Language Model (LLM) Basics

Transformers with billions of parameters trained on large corpora, i.e., large language models (LLMs), have demonstrated an unprecedented understanding of natural language and good logical reasoning ability based on factual knowledge [34]. Based on the part of transformer utilized for language modeling, existing LLMs can be categorized into three classes: encoder-only LLMs, such as BERT [14], encoder-decoder-based LLMs, such as T5 [25], and decoder-only LLMs, such as GPT [24] and LlaMA [27], etc. We focus on LLMs with decoders due to their superior generative abilities compared with the encoder-only models [22]. The training of LLMs is mainly based on two stages. In the pretraining stage, LLMs are trained on large corpora such as website content, Wikipedia, ArXiv paper, and GitHub codes via language modeling (i.e., next/masked token prediction), where knowledge in the corpus can be effectively encoded in the weights of the transformer network facilitated by the stacked self-attention modules. Then, during the finetuning stage, exemplar prompt-output pairs (such as questions and answers) or human feedback on multiple generated answers are provided to the LLMs such that they can conduct logical reasoning and generate answers based on the encoded knowledge from the pretrained stage.

### 2.2 LLM in Recommender Systems

Recently, LLM-based RS has attracted extensive attention from both academia and industry, which are promising to address the long-standing issues of traditional ID-based RS, such as shallow textual information understanding, poor generalization, etc. [18, 19]. Hou et al. [9] show that existing LLMs can be viewed as zero-shot rankers, which can rank the relevance of movies based on user historical interactions and movie descriptions. However, since pretrained LLMs are not aligned with the recommendation task,

more efforts have been devoted to the finetuning of LLMs to obtain recommendation-oriented models. An exemplar work is P5 [7], which finetunes T5 with token sequences transformed from interactions and user/item features, where items are presented by pseudo-IDs in the form of "item_i". Afterwards, M6 [4] was proposed that combines text infilling and auto-regression in the pretraining stage, where pseudo IDs in P5 are completely avoided and replaced by textual descriptions. Recently, TALLRec [1] was proposed where items are represented by both pseudo-ID and textual descriptions. Pseudo-ID-based item representations can easily introduce spurious correlations between irrelevant items. To address this issue, Hua et al. [11] proposed to introduce a small number of new tokens, where tokens used to describe the items are determined by their content and collaborative similarity. However, representing items with multiple shared tokens can still introduce bias. In addition, for the above methods, candidate items need to be explicitly provided in the prompt when conducting direct recommendation, where the size of candidate pool is limited. Finally, recommendations are generated via autoregression, which is highly inefficient. In summary, the dichotomy between natural language and RS is not well addressed in the current explorations of LLM-based RSs, leading to unsatisfactory user/item representations, ineffective collaborative/content modeling, and low efficiency in generating recommendations.

## 3 METHODOLOGY

### 3.1 Problem Formulation

In this paper, we focus on recommendations with implicit feedback [10]. Consider a system of $I$ users and $J$ items. We use a binary rating vector $\mathbf{r}_i \in \{0, 1\}^J$ to denote whether user $i$ has interacted with the $J$ items. In addition, we use $\mathbf{x}_i^u$, $\mathbf{x}_j^v$ to denote the textual features associated with user $i$ and item $j$, such as user biography and item content, etc. $\mathbf{x}_{ij}^{uv}$ denotes the textual features associated with both user $i$ and item $j$, such as user $i$'s review for item $j$. Hereafter, we take a sequential view of $\mathbf{x}_{\{i,j,ij\}}^{\{u,v,uv\}}$, where $\mathbf{x}_{\{i,j,ij\},k}^{\{u,v,uv\}}$ is a size $N$ one-hot vector denoting the $k$th token in the textual sequence[2]. In addition, we have a pretrained large language model (LLM), of which we take a probabilistic view and denote it as $p_{llm}(\mathbf{x}_{k+1}|\mathbf{x}_{1:k})$, which transform $\mathbf{x}_{1:k}$ into a latent sequence $\mathbf{h}_{1:k}^{(L)} \in \mathbb{R}^{k \times K_h}$ via $L$ stacked self-attention modules $llm(\mathbf{x}_{1:k})$ and maps the $\mathbf{h}_k^{(L)}$ to the probability space of the next token $\mathbf{x}_{k+1}$. Since the LLM is pretrained on large corpora and finetuned on exemplar prompt-answer pairs, the generation is based on logical reasoning with the context information in $\mathbf{x}_{1:k}$ according to its pretrained knowledge.

Our aim is to design a new RS that tightly couples the LLM with the recommendation task by introducing user/item ID tokens (and token embeddings), such that user/item semantics (e.g., user interests in item) can be accurately modeled for effective and efficient recommendation whereas the encoded knowledge and reasoning ability of the pretrained LLMs can be fully utilized simultaneously.

### 3.2 Extension of User/Item Tokens

#### 3.2.1 *Vocab Expansion*. To tightly couple the pretrained LLM with the recommendation task, we first expand the vocabulary of

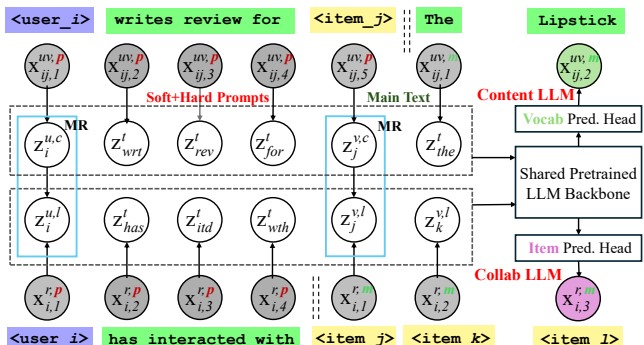

**Figure 2: The overview of the proposed CLLM4Rec in the mutually-regularized pretraining stage. Mutual regularization of item_k is omitted for simplicity.**

the LLM by adding user/item ID tokens to describe the intrinsic user/item semantic, such that semantic gap between RS and natural language can be well bridged. We use bracket notations **"<user_i>"** and **"<item_j>"** to denote the newly-introduced token for the $i$th user and the $j$th item, respectively, which has token ID $N + i$ and $N + I + j$, and will not be broken down into atomic tokens.

#### 3.2.2 *Token Embeddings*. For LLMs to understand the tokens, they must be first transformed into dense embeddings. Accordingly, we use $\mathbf{z}_k^t \in R^K$ to represent the pretrained embedding of the $k$th vocab token. In addition, for the newly-introduced user/item tokens, we introduce two types of embeddings to represent user/item collaborative and content semantics. Specifically, to align the user/item tokens with the vocab space of the pretrained LLM, we sample the user/item collaborative token embeddings from the same size-$K$ latent space as follows:

$$\mathbf{z}_i^{l,u}, \mathbf{z}_j^{l,v} \sim \mathcal{N}\left(\mathbf{0}, \lambda_l^{-1} \cdot \mathbf{I}_K\right), \tag{1}$$

where $\lambda_l$ is the prior precision for $\mathbf{z}_i^{l,u}, \mathbf{z}_j^{l,v}$. Importantly, to align the content semantics with the collaborative semantic for more recommendation-oriented content modeling, we sample the user/item content token embeddings from the following conditional prior:

$$\mathbf{z}_i^{c,u} \sim \mathcal{N}\left(\mathbf{z}_i^{l,u}, \lambda_c^{-1} \cdot \mathbf{I}_K\right), \mathbf{z}_j^{c,v} \sim \mathcal{N}\left(\mathbf{z}_j^{l,v}, \lambda_c^{-1} \cdot \mathbf{I}_K\right). \tag{2}$$

where $\lambda_c$ is the precision for the conditional prior of $\mathbf{z}_i^{c,u}, \mathbf{z}_j^{c,v}$. The horizontally-stacked matrices of vocab/collaborative/content token embeddings are denoted as $\mathbf{Z}^t$, $\mathbf{Z}^{l,\{u,v\}}$, and $\mathbf{Z}^{c,\{u,v\}}$, respectively[3].

#### 3.2.3 *CLLM4Rec Base Model*. With user/item tokens and the corresponding token embeddings introduced in the previous subsections, we are ready to introduce the CLLM4Rec base model with expanded vocabulary. The CLLM4Rec base model is denoted with

$$\mathbf{h}_{\{l,c\},1:k}^{(L)} = \hat{llm}_{\{l,c\}}(\mathbf{x}_{1:k}), \tag{3}$$

which maps the token sequence $\mathbf{x}_{1:k}$ into the hidden space $\mathbf{h}_{\{l,c\},1:k}^{(L)}$ through $L$ stacked self-attention module (the superscript $(L)$ will be omitted if no ambiguity exists); here, $\mathbf{x}_k$ is a size $N + I + J$ one-hot

**User ID: 0057  Item ID: 0046**

**Item Title:** `Wet n Wild Mega Last`
`Lip Color 908C Sugar Plum Fairy`

**Review:** `The color is a perfect mix of`
`dark purple, red and pink. The only`
`downside is the drying aspect of the`
`lipstick, which I counteract by using`
`lip balm before putting it on.`

**Figure 3: Example review data from Amazon Beauty dataset.**

vector denoting the token of either a vocab, a user, or an item. In addition, the subscript in $l\hat{l}m_{\{l,c\}}$ denotes which embedding matrix is used to encode the user/item tokens (where $l$ stands for matrix $\mathbf{Z}^{l,\{u,v\}}$ and $c$ stands for matrix $\mathbf{Z}^{c,\{u,v\}}$). For the CLLM4Rec base model $l\hat{l}m_{\{l,c\}}$, only the user/item token embeddings are trainable, whereas the vocab embeddings $\mathbf{Z}^t$ as well as the other parts of the backbone LLM are fixed to preserve the pretrained knowledge.

### 3.3 Mutually-Regularized Pretraining

With CLLM4Rec base model introduced in the previous section, we discuss the mutually-regularized pretraining strategy for CLLM4Rec to learn the user/item collaborative/content token embeddings based on language modeling on corpora established from user-item interactions and user/item textual features, where the encoded knowledge and logical reasoning ability of the pretrained LLM can be fully utilized. The overall process can be referred to in Fig. 2.

*3.3.1 **Recommendation-Specific Corpora**.* Generally, we can transform the interactions and user/item content features into documents of user/item/vocab token sequences as follows:

---

**Raw Corpora Transformed from Recommendation Data**

**(a)** *Historical Interactions* $\mathbf{r}_i$:

`<user_i>` `has interacted with` `<item_j> <item_k> ...`

**(b)** *User/Item Textual Features* $\mathbf{x}_i^u, \mathbf{x}_j^v, \mathbf{x}_{ij}^{uv}$:

`The biography of` `<user_i>` `is: Main biography.`
`The content of` `<item_j>` `is: Main contents.`
`<user_i>` `writes the review for` `<item_j>` `: Main reviews.`

---

where an example based on the Amazon Beauty dataset can be referred to in Fig. 3. However, directly conducting language modeling on the raw corpora is clearly infeasible, as each document is composed of heterogeneous vocab, user, and item tokens, where the number of meaningful vocab tokens (e.g., ∼ 50k for GPT, and ∼ 30k for T5) can be diluted by the large number of newly introduced user/item tokens with randomly initialized embeddings.

*3.3.2 **Soft+Hard Prompting**.* To address the above challenge, we propose a novel soft+hard prompting strategy to facilitate language modeling on RS-specific corpora with heterogeneous user/item/vocab tokens. The strategy is based on a key observation that documents transformed from both user-item interactions $\mathbf{r}_i$ and user/item textual features $\mathbf{x}_i^u, \mathbf{x}_j^v, \mathbf{x}_{ij}^{uv}$ can be broken down into two parts: A heterogeneous part composed of soft (user/item) and hard (vocab) tokens providing context information regarding the gist of the document, and a main text part with homogeneous item/vocab tokens

filling the pretexts in detail. Therefore, we can view the first part as a soft+hard prompt and conduct language modeling only on the main text. This encourages the model to focus exclusively on collaborative and content information, such that the stability and effectiveness of language modeling can be substantially enhanced.

For example, the document $\mathbf{x}_i^r$ transformed from the historical interactions of user $i$ can be broken down into the soft+hard prompt $\mathbf{x}_i^{r,p}$ and homogeneous item token sequence $\mathbf{x}_i^{r,m}$ as follows[4]:

---

**(a)** *Historical Interactions* $\mathbf{r}_i$:

`<user_i>` `has interacted with` `<item_j> <item_k> ...`

$\underbrace{\qquad\qquad}_{\text{soft+hard prompt } \mathbf{x}_i^{r,p}} \quad \underbrace{\qquad\qquad}_{\text{item token seq. } \mathbf{x}_i^{r,m}}$

---

Accordingly, we introduce the **collaborative LLM** by adding an item prediction head $f_l : \mathbb{R}^{K_h} \to \mathbb{P}(J)$ to the CLLM4Rec base model $l\hat{l}m_l$, which maps the final-layer last-step hidden representation $\mathbf{h}_{l,-1}$ calculated via $l\hat{l}m_l$ to the item probability space $\mathbb{P}(J)$ to predict the next item token. The weights of $f_l$ are tied with the item collaborative token embeddings $\mathbf{Z}^{l,v}$ as $f_l(\mathbf{h}_{l,-1}) = \text{softmax}(\mathbf{Z}^{l,v} \cdot \mathbf{h}_{l,-1})$. The generative process of the collaborative LLM can be denoted as:

$$\mathbf{x}_{i,k+1}^{r,m} \sim p_{l\hat{l}m_l}^{f_l} \left( \mathbf{x}_{i,k+1}^{r,m} | \mathbf{x}_{i,1:k}^{r,m}, \mathbf{x}_i^{r,p} \right), \qquad (4)$$

where the prompt $\mathbf{x}_i^{r,p}$ serves as a context to generate the next item token based on previous item tokens. Since the generation of $\mathbf{x}_{i,k+1}^{r,m}$ requires attending to previous tokens, when maximizing the likelihood, the collaborative LLM pushes the token embeddings of user $i$, i.e., $\mathbf{z}_i^{l,u}$, and the token embeddings of the interacted items, i.e., $\mathbf{z}_j^{l,v}, \mathbf{z}_k^{l,v}, \cdots$, to be close to each other, where user/item collaborative semantics in recommendation can be accurately captured.

Similarly, for the documents transformed from the user/item content[5], it can also naturally be split into a soft+hard prompt $\mathbf{x}_{ij}^{uv,p}$ and the main text $\mathbf{x}_{ij}^{uv,m}$ of homogeneous vocab token sequence as:

---

**(b)** *User/Item Textual Features* $\mathbf{x}_{ij}^{uv}$:

`<user_i>` `writes the review for` `<item_j>` `: Main reviews.`

$\underbrace{\qquad\qquad}_{\text{soft+hard prompt } \mathbf{x}_{ij}^{uv,p}} \quad \underbrace{\qquad\qquad}_{\text{vocab seq. } \mathbf{x}_{ij}^{uv,m}}$

---

Accordingly, we introduce the **content LLM** by adding a vocab prediction head $f_c : \mathbb{R}^{K_h} \to \mathbb{P}(N)$ to the CLLM4Rec base model $l\hat{l}m_c$, which maps the final-layer last-step hidden representation $\mathbf{h}_{c,-1}$ calculated via $l\hat{l}m_c$ (which shares the same pretrained LLM with $l\hat{l}m_l$ but uses $\mathbf{Z}^{c,\{u,v\}}$ to decode the user/item token) to the vocab probability space. Similarly, the weights of $f_c$ are tied with the vocab embeddings $\mathbf{Z}^t$ as $f_c(\mathbf{h}_{c,-1}) = \text{softmax}(\mathbf{Z}^t \cdot \mathbf{h}_{c,-1})$. The generative process of the content LLM can be denoted as follows:

$$\mathbf{x}_{ij,k+1}^{c,m} \sim p_{l\hat{l}m_c}^{f_c} \left( \mathbf{x}_{ij,k+1}^{uv,m} | \mathbf{x}_{ij,1:k}^{uv,m}, \mathbf{x}_{ij}^{uv,p} \right), \qquad (5)$$

which generates the next vocab token $\mathbf{x}_{ij,k+1}^{uv,m}$ based on previously generated vocab tokens $\mathbf{x}_{ij,1:k}^{uv,m}$ with prompt $\mathbf{x}_{ij}^{uv,p}$ as the context.

---

[4]We use the superscripts $p$ and $m$ to distinguish the prompt and the main text.
[5]Hereafter, we take $\mathbf{x}_{ij}^{uv}$, i.e., the textual feature associated with user $i$ and item $j$ as an example for discussions, which can be easily generalized to the case of $\mathbf{x}_i^u$ and $\mathbf{x}_j^v$

When maximizing the likelihood, the content information in $\mathbf{x}^{uv,m}$ can be encoded in the content token embeddings of user $i$ and item $j$, i.e., $\mathbf{z}_i^{c,u}$, $\mathbf{z}_j^{c,v}$, where the pretrained knowledge of the LLM can be fully utilized. For example, for the reviews shown in Fig. 3, the pretrained LLM will know that **<item_46>** is a lipstick with dark purple, red, and pink colors and can have side effects of drying lip, and reasons that **<user_57>** likes the colors but hates the side effects, which can be alleviated by the lip balm.

**Discussion.** Generally, since the "hard" (i.e., the vocab) part of the prompts $\mathbf{x}_i^{r,p}$ and $\mathbf{x}_{ij}^{uv,p}$ is what the pretrained LLM could understand, they are designed to trigger the reasoning ability of the pretrained LLM based on its encoded knowledge. For example, the relational phrase **"has interacted with"** in the prompt $\mathbf{x}_i^{r,p}$ guides the collaborative LLM to understand that the newly-introduced token **<user_i>** is a *user subject* and the tokens in the prompt $\mathbf{x}_i^{r,m}$ are the *objects* of interacted item sequences. Meanwhile, the contexts **"write the review for"** in $\mathbf{x}_{ij}^{uv,p}$ direct the content LLM to better understand the nature of main texts in $\mathbf{x}_{ij}^{uv,m}$, i.e., **<user_i>**'s judgment on the **<item_j>** based on the personal using experience. The specific formulation of the prompt can be flexible, as Geng et al. [7] has demonstrated that the variation in the expression of the prompt makes less difference, as long as the meaning is the same and the prompt is consistent across the training and testing phases.

### 3.3.3 *Mutually-Regularization*.

Since the pretrained LLMs are not recommendation-oriented, naively optimizing the language modeling objective as Eq. (5) unavoidably captures noise irrelevant to recommendations. In addition, since the user/item interactions are sparse, the collaborative LLM can easily overfit on the observed interactions. To address this issue, we propose the mutually-regularized pretraining for CLLM4Rec, where collaborative LLM can guide content LLM to capture recommendation-oriented information from user/item content, and content LLM can in turn introduce side information to support collaborative filtering.

The mutual-regularization naturally arises with the generative process of the CLLM4Rec pretraining stage defined in the previous subsections. If we denote the stacked item token embeddings as $\mathbf{Z}_i^{c,v}$, $\mathbf{Z}_i^{l,v}$, which contains item $j$ and other items interacted by the user $i$, the generation process of CLLM4Rec associated with $\mathbf{x}_i^r$ and $\mathbf{x}_{ij}^{uv}$ can be defined as the joint distribution as follows:

$$p\left(\mathbf{x}_i^{r,m}, \mathbf{x}_{ij}^{uv,m}, \mathbf{z}_i^{l,u}, \mathbf{Z}_i^{l,v}, \mathbf{z}_i^{c,u}, \mathbf{Z}_i^{c,v} \middle| \mathbf{x}_i^{r,p}, \mathbf{x}_{ij}^{uv,p}\right) =$$

$$\underbrace{\Pi_k p_{ll\hat{m}_l}^{f_l}\left(\mathbf{x}_{i,k}^{r,m} \middle| \mathbf{x}_{i,1:k-1}^{r,m}, \mathbf{x}_i^{r,p}\right)}_{\text{LM for collab. LLM}} \cdot \underbrace{\Pi_k p_{ll\hat{m}_c}^{f_c}\left(\mathbf{x}_{ij,k}^{uv,m} \middle| \mathbf{x}_{ij,1:k-1}^{uv,m}, \mathbf{x}_i^{uv,p}\right)}_{\text{LM for content LLM}} \cdot$$

$$\underbrace{p\left(\mathbf{z}_i^{c,u} \middle| \mathbf{z}_i^{l,u}\right) \cdot \Pi_k p\left(\mathbf{z}_{ik}^{c,v} \middle| \mathbf{z}_{ik}^{l,v}\right)}_{\text{mutual regularization}} \cdot \underbrace{p\left(\mathbf{z}_i^{l,u}\right) \cdot \Pi_k p\left(\mathbf{z}_{ik}^{l,v}\right)}_{\text{prior}}.$$

(6)

A scrutiny of Eq. (6) reveals that the joint distribution can be decomposed into three parts: **1)** the language modeling of the collaborative and content LLMs that learn user/item token embeddings as Eqs. (4) and (5); **2)** the mutual regularization that connects the user/item token embeddings of the two LLMs (i.e., according to Eqs. (1-2),

$p\left(\mathbf{z}_i^{c,u} \middle| \mathbf{z}_i^{l,u}\right)$ and $p\left(\mathbf{z}_{ik}^{c,v} \middle| \mathbf{z}_{ik}^{l,v}\right)$ are conditional Gaussians, which will introduce MSE regularization between $\mathbf{z}_i^{c,u}$, $\mathbf{z}_i^{l,u}$, and $\mathbf{z}_{ik}^{c,v}$, $\mathbf{z}_{ik}^{l,v}$ when log-likelihood is maximized) **3)** the prior of $\mathbf{z}_i^{l,u}$ and $\mathbf{z}_{ik}^{l,v}$, which will be ignored due to the existence of mutual regularization (i.e., setting the precision $\lambda_l$ in the prior in Eq. (1) as zero).

We use Maximum a Posteriori (MAP) to estimate the user/item token embeddings $\mathbf{z}_i^{l,u}, \mathbf{Z}_i^{l,v}, \mathbf{z}_i^{c,u}, \mathbf{Z}_i^{c,v}$, where the objective is proportional to the logarithm of the joint distribution specified in Eq. (4). We take alternative steps to optimize the MAP objective. If we denote the trainable parameters associated with the item token prediction head $f_l$ and vocab token prediction head $f_c$ as $\theta_l$ (which are tied with the corresponding token embeddings), the objective for the collaborative LLM (L-step) and content LLM (C-step) with mutual regularization can be derived as follows:

**L-step.** In the L-step, we fix user/item content embeddings $\mathbf{z}_i^{c,u}$, $\mathbf{Z}_i^{c,v}$ as $\hat{\mathbf{z}}_i^{c,u}$, $\hat{\mathbf{Z}}_i^{c,v}$ in Eq. (6), and use them to constrain the user/item collaborative embeddings along with the language modeling of collaborative LLM, leading to the following composite objective:

$$\mathcal{L}_{\text{l\_step}}^{\text{MAP}}\left(\mathbf{z}_i^{l,u}, \mathbf{Z}_i^{l,v}, \theta\right) = -\sum_k \underbrace{\ln p_{ll\hat{m}_l}^{f_l}\left(\mathbf{x}_{i,k}^{r,m} \middle| \mathbf{x}_{i,1:k-1}^{r,m}, \mathbf{x}_i^{r,p}\right)}_{\text{LM loss for collab. LLM}}$$

$$\underbrace{-\frac{\lambda_c}{2}\left\|\mathbf{z}_i^{l,u} - \hat{\mathbf{z}}_i^{c,u}\right\|_2^2 - \sum_k \frac{\lambda_c}{2}\cdot\left\|\mathbf{z}_{ik}^{l,v} - \hat{\mathbf{z}}_{ik}^{c,v}\right\|_2^2}_{\text{MR loss with content LLM}} \underbrace{- \frac{\lambda_l}{2}\left\|\mathbf{z}_i^{l,u}\right\| - \frac{\lambda_l}{2}\left\|\mathbf{z}_j^{l,v}\right\|}_{\text{Prior loss}} + C_l,$$

(7)

where $C_l$ is the constant irrelevant for optimization. The **_LM loss_** captures the collaborative similarity between token embeddings of user $i$ and the interacted items, where side information can be introduced via the **_MR loss_** to support collaborative filtering.

**C-step.** After one-step optimization of the L-step, we fix the user/item collaborative token embeddings $\mathbf{z}_i^{l,u}, \mathbf{z}_j^{l,v}$ as $\hat{\mathbf{z}}_i^{l,u}, \hat{\mathbf{z}}_j^{l,v}$ in Eq. (6), leading to the following composite objective for the content LLM:

$$\mathcal{L}_{\text{c\_step}}^{\text{MAP}}\left(\mathbf{z}_i^{c,u}, \mathbf{z}_j^{c,v}, \theta\right) = -\sum_k \underbrace{\ln p_{ll\hat{m}_c}^{f_c}\left(\mathbf{x}_{ij,k}^{uv,m} \middle| \mathbf{x}_{ij,1:k-1}^{uv,m}, \mathbf{x}_i^{uv,p}\right)}_{\text{LM loss for content LLM}}$$

$$\underbrace{-\frac{\lambda_c}{2}\left\|\mathbf{z}_i^{c,u} - \hat{\mathbf{z}}_i^{l,u}\right\|_2^2 - \frac{\lambda_c}{2}\cdot\left\|\mathbf{z}_j^{c,v} - \hat{\mathbf{z}}_j^{l,v}\right\|_2^2}_{\text{MR loss with collab. LLM}} + C_c,$$

(8)

where MR loss constrains content LLM to capture recommendation-oriented information from user/item textual features. In Eqs. (7) and (8), $\lambda_c$ controls the strength of mutual regularization, which will be thoroughly discussed in the empirical study.

### 3.3.4 *Stochastic Item Reordering*.

Another issue that hinders effective collaborative filtering via Eq. (7) is the order of item tokens when transforming the historical interactions $\mathbf{r}_i$ into a token sequence $\mathbf{x}_i^{r,m}$ for language modeling. Item order usually does not matter for collaborative filtering (even if it matters, the positional embeddings denoting the order of natural language may not capture the semantics of the order of interactions). To address this issue, we propose to randomly permute the item tokens in $\mathbf{x}_i^{r,m}$

with prompt $\mathbf{x}_i^{r,p}$ fixed when optimizing the collaborative LLM as Eq. (7). Through this strategy, the order of interacted items can be ignored without negative influence on the vocab tokens in $\mathbf{x}_i^{r,p}$.

## 3.4 Recommendation-Oriented Finetuning

*3.4.1 **Pretraining v.s. Finetuning**.* The pretraining of CLLM4Rec aims to learn user/item token embeddings based on the large corpus of documents transformed from user-item interactions $\mathbf{r}_i$ and user/item textual features $\mathbf{x}_i^u, \mathbf{x}_j^v, \mathbf{x}_{ij}^{uv}$ via language modeling. However, for now, the pretrained CLLM4Rec can only complete item/vocab token sequences based on the soft+hard prompts, and therefore the gap between NLP and RS is still not completely eliminated. In addition, naively treating the collaborative LLM as a recommendation model can lead to huge computational costs where the recommended items are sequentially generated via auto-regression. Therefore, we propose a recommendation-oriented finetuning strategy for CLLM4Rec, which aims to finetune the pretrained collaborative LLM and tailor it for efficient recommendations.

*3.4.2 **Masked Prompting with Multinomial Head**.* To achieve this purpose, we first design a masked prompting strategy to generate recommendation-oriented prompts. For each user, we randomly mask the interacted items $\mathbf{r}^i$ by $100 \times p_m\%$, where the remaining items are denoted as $\mathbf{r}_i^{masked}$, and use it to generate a recommendation-oriented prompt $\mathbf{x}_i^{rec,p}$. All the hold-out items, which we denote with a multi-hot vector $\mathbf{r}_i^{hold}$, are treated as the target. The prompt $\mathbf{x}_i^{rec,p}$ based on $\mathbf{r}_i^{masked}$ is designed as:

> **(c) Recommendation Prompts & Target**
> (prompt) <user_i> has interacted with <item_j'> <item_k'>
> the user will interact with: (target) $\mathbf{r}_i^{hold}$,

which triggers the reasoning ability of the pretrained LLM by using relational phrase **"has interacted with"** to describe the historical interactions, and using the phrase **"the user will interact with"** to usher into the prediction of the target items $\mathbf{r}_i^{hold}$.

We name CLLM4Rec in the finetuning stage as **RecLLM**, which inherits the CLLM4Rec base model $\hat{llm}_l$ from the collaborative LLM in the pretraining stage and introduces a new item prediction head with multinomial likelihood, i.e., $f_{rec}$, whose weights are also tied with the item token embeddings $\mathbf{Z}^{l,v}$. The generation of the hold-out items $\mathbf{r}_i^{hold}$ via the RecLLM can be formulated as follows:

$$\mathbf{r}_i^{hold} \sim multi\left(f_{rec}\left(\mathbf{h}_{l,i,-1}^{rec}\right), N_i^{hold}\right), \text{ where } \mathbf{h}_{l,i}^{rec} = \hat{llm}_l\left(\mathbf{x}_i^{rec,p}\right), \tag{9}$$

where $multi$ denotes the multinomial distribution and $N_i^{hold}$ is the number of hold-out items for user $i$. When finetuning the RecLLM according to Eq. (9), $\mathbf{h}_{l,i,-1}^{rec}$, which can be viewed as the user latent variable summarizing the historical interaction of user $i$, is encouraged to be similar to the collaborative embeddings of all the interacted items. In addition, we keep it regularized with the content LLM in a similar manner as Eq. (7), and use the stochastic item reordering strategy to generate the prompt $\mathbf{x}_i^{rec,p}$ [6]. Through the proposed finetuning strategy, CLLM4Rec can fully utilize the encoded knowledge from the pretrained LLM backbone and the

---

[6] The objective of the RecLLM is formulated in Eq. (10) in Appendix A.2.

user/item token embeddings learned from the mutually-regularized pretraining stage to efficiently generate recommendations in a single forward-propagation step, where all $J$ items serve as candidates.

## 3.5 Predictions with CLLM4Rec

After the pretraining and finetuning of CLLM4Rec, to make recommendation for user $i$, we can convert the *whole* historical interactions of the user, i.e., $\mathbf{r}_i$, into the recommendation-oriented prompt $\hat{\mathbf{x}}_i^{rec,p}$ as described in Section 3.4.2 (with no masked items) and input it into the RecLLM model. Then, the multinomial probability $\hat{\mathbf{r}}_i$ **over all $J$ items** can be obtained through one forward propagation via $\hat{\mathbf{r}}_i = multi\left(f_{rec}\left(\hat{\mathbf{h}}_{i,-1}^{rec}\right)\right), \hat{\mathbf{h}}_i^{rec} = \hat{llm}_l\left(\hat{\mathbf{x}}_i^{rec,p}\right)$, where uninteracted items with top-$M$ scores in $\hat{\mathbf{r}}_i$ can be selected as recommendations.

## 4 EMPIRICAL STUDY

In this section, we present the experiments on four public datasets and one Company dataset[7] to demonstrate the effectiveness of CLLM4Rec, aiming to answer the following research questions.

- **RQ1.** How does CLLM4Rec, the first RS that tightly couples the ID-based paradigm with the LLM-based paradigm, perform compared to state-of-the-art ID-based and LLM-based RSs?
- **RQ2.** How does the **pretraining stage** of CLLM4Rec (including the mutual regularization trick and the stochastic item reorder strategy) influence the performance of CLLM4Rec?
- **RQ3.** How does the **finetuning stage** of CLLM4Rec with masked prompt and multinomial item prediction head influence the efficiency and effectiveness of recommendations.

## 4.1 Experimental Setup

*4.1.1 **Datasets**.* The experiments are mainly based on four public datasets: Amazon (AM)-Beauty dataset, AM-Toys dataset, AM-Sports dataset [21] and the Yelp dataset [35], where we binarize the interactions by keeping only ratings > 3 and treat them as implicit feedback [17]. In addition, we filter the dataset such that they keep the original 5-core property after binarization. For each user, we randomly select 80% of interactions for training, 10% for validation, and 10% for testing, where as least one item is selected in the validation and the test set. The reviews that users provide to the items are collected as the textual feature $\mathbf{x}_{ij}^{uv}$. The **real-world experiments** are based on a job recommendation dataset collected nearline at the Company, where user's click on the job Ads are logged as the implicit feedback, and users' self-provided biography $\mathbf{x}_i^u$ and the job descriptions $\mathbf{x}_j^v$ are collected as the textual features, respectively. The statistics of the dataset are summarized in Table 3 in Appendix.

*4.1.2 **Implementation Details**.* Due to the space limitation, we only discuss CLLM4Rec with GPT-2 backbone with token embedding 768 and token size 50,257 in this section, where experiments with T5 backbone are discussed in Appendix B. During the training stage, we first optimize the content LLM as Eq. (5) via language modeling for 10 epochs to warm up the user/item content token embeddings. Then, in the mutually-regularized pretraining stage, we alternatively train the collaborative and content LLMs as specified in Eqs. (7) and (8) for 100 epochs. Finally, we conduct

---

[7] Company name omitted according to the double-blind review policy.

the recommendation-oriented finetuning for 150 epochs, where the RecLLM is monitored with metrics Recall@20, Recall@40, and NDCG@100 calculated on the validation set as with [17]. RecLLM with the best performance are logged and evaluated on the test set as the final results. $\lambda_c$ in Eqs. (7) and (8) is an important hyperparameter, we first fix its value to the optimal one found by grid search, and then discuss its influence in Section 4.3.

## 4.2 Comparison with Baselines

*4.2.1 Baselines.* To demonstrate the multifaceted superiority of the proposed CLLM4Rec, we include the following ID-based and (L)LM-based RSs as the baselines for comparisons:

**ID-based Baselines.**

- **Multi-Vae** [17] is an ID-based collaborative filtering baseline that recommends new items by reconstructing the ratings $\mathbf{r}_i$ via a variational auto-encoder (VAE) with multinomial likelihood.
- **Md-Cvae** [36] is a hybrid RS that extends the Multi-VAE by introducing a dual feature VAE on textual features $\mathbf{x}_{ij}^{uv}$ to regularize the reconstruction of $\mathbf{r}_i$ in the Multi-VAE.

**LM-based Baselines**[8].

- **Bert4Rec** [26] uses masked language modeling (MLM) proposed in BERT [14] to learn user/item embeddings for recommendation with bidirectional self-attention mechanism.
- **S³Rec** [35] extends BERT4Rec by augmenting the MLM with auxiliary tasks such as item attribute prediction, where content features can be fused for self-supervised learning.

**LLM-based Baselines.**
**(a) Qualitative Analysis.**
    Both pseudo-ID-based and description-based methods discussed in Section 2.2 represent user/item with multiple tokens and formulate direct recommendation as a token generation problem. Since the generated tokens could be irrelevant to the recommendation purpose, candidate items usually need to be explicitly provided in the prompt (e.g., P5 [7] provides 100 candidate items where one is positive, and TALLRec [1] outputs yes/no decision based on user/item descriptions in the prompts, etc.). In contrast, CLLM4Rec can generate multiple recommendations from the entire candidate pool. Therefore, these methods cannot directly work in our setting, and the comparisons are mainly based on qualitative analysis.
**(b) Quantitative Analysis**
    In addition, we design the following LLM-based baselines to quantitatively demonstrate the effectiveness of CLLM4Rec.

- **Llm-Scratch** has the same structure as CLLM4Rec, but it trains the whole model from scratch instead of loading and fixing the weights of the pretrained LLM backbone.
- **Llm-CF** eliminates the content LLM from CLLM4Rec and the mutually-regularized pretraining step and uses only the collaborative LLM and RecLLM for recommendation.
- **Llm-FTALL** has the same structure as CLLM4Rec, but it finetunes the whole network including the vocab embeddings as well as other parts of the pretrained LLM, instead of training only the newly-introduced user/item token embeddings.

---

[8]Note that both Bert4Rec and S³Rec are original designed for sequential recommendation. In this paper, we use similar recommendation-oriented finetuning as CLLM4Rec to adapt them to direct recommendation, where item sequences generated from masked interactions are used to predict all hold-out items with multinomial likelihood.

**Table 1: Comparison between CLLM4Rec and various baselines with GPT-backbone on three Amazon Review datasets.**

| AM-Beauty | Recall@20 | Recall@40 | NDCG@100 |
|---|---|---|---|
| Multi-VAE | 0.1295 | 0.1720 | 0.0835 |
| MD-CVAE | 0.1472 | 0.2058 | 0.0976 |
| BERT4Rec | 0.1126 | 0.1677 | 0.0781 |
| S³Rec | 0.1354 | 0.1789 | 0.0867 |
| LLM-Scratch | 0.0840 | 0.1265 | 0.0583 |
| LLM-CF | 0.1319 | 0.1841 | 0.0855 |
| LLM-FtAll | 0.1335 | 0.1984 | 0.0836 |
| LLM-FixOrd | 0.1524 | 0.2219 | 0.1072 |
| LLM-PreRec | 0.1547 | 0.2196 | 0.1051 |
| **CLLM4Rec** | **0.1656** | **0.2323** | **0.1118** |

| AM-Toys | Recall@20 | Recall@40 | NDCG@100 |
|---|---|---|---|
| Multi-VAE | 0.1076 | 0.1558 | 0.0781 |
| MD-CVAE | 0.1291 | 0.1804 | 0.0844 |
| BERT4Rec | 0.0853 | 0.1375 | 0.0532 |
| S³Rec | 0.1064 | 0.1524 | 0.0665 |
| LLM-Scratch | 0.0485 | 0.0771 | 0.0362 |
| LLM-CF | 0.1027 | 0.1434 | 0.0680 |
| LLM-FtAll | 0.1162 | 0.1542 | 0.0696 |
| LLM-FixOrd | 0.1342 | 0.1887 | 0.0889 |
| LLM-PreRec | 0.1308 | 0.1859 | 0.0874 |
| **CLLM4Rec** | **0.1436** | **0.1933** | **0.0918** |

| AM-Sports | Recall@20 | Recall@40 | NDCG@100 |
|---|---|---|---|
| Multi-VAE | 0.0659 | 0.0975 | 0.0446 |
| MD-CVAE | 0.0714 | 0.1180 | 0.0514 |
| BERT4Rec | 0.0521 | 0.0701 | 0.0305 |
| S³Rec | 0.0616 | 0.0813 | 0.0438 |
| LLM-Scratch | 0.0362 | 0.0538 | 0.0362 |
| LLM-CF | 0.0642 | 0.0966 | 0.0419 |
| LLM-FtAll | 0.0794 | 0.1002 | 0.0424 |
| LLM-FixOrd | 0.0901 | 0.1295 | 0.0592 |
| LLM-PreRec | 0.0839 | 0.1248 | 0.0561 |
| **CLLM4Rec** | **0.0926** | **0.1351** | **0.0634** |

- **Llm-FixOrd** has the same structure as CLLM4Rec but it removes the stochastic item reordering strategy for both the collaborative LLM in pretraining and the RecLLM in finetuning.
- **Llm-PreRec** discards finetuning and ranks the categorical probability from the next item token prediction head of the collaborative LLM in the pretraining stage to make recommendations.

*4.2.2 Results on the Public Datasets.* We first analyze the experimental results on four public datasets to provide preliminary answers for **RQs. 1, 2, 3**. From Tables 1 and 2, we can find that the ID-base method, Multi-VAE, remains a strong baseline for collaborative filtering (CF). LLM-CF, the CF backbone of CLLM4Rec, cannot beat Multi-VAE on both AM-Sports and Toys datasets, even if the "hard" part of the prompt triggers the reasoning ability of the pretrained LLM. However, when large textual data are available, CLLM4Rec outperforms its ID-based counterpart, MD-CVAE (which tightly couples an item content VAE with the Multi-VAE)

**Table 2: Comparison between CLLM4Rec and various baselines on the Yelp dataset and the Company dataset.**

| Yelp | Recall@20 | Recall@40 | NDCG@100 |
|---|---|---|---|
| Multi-VAE | 0.0526 | 0.0842 | 0.0424 |
| MD-CVAE | 0.0664 | 0.1058 | 0.0497 |
| BERT4Rec | 0.0418 | 0.0724 | 0.0361 |
| SASRec | 0.0563 | 0.0893 | 0.0485 |
| LLM-Scratch | 0.0199 | 0.0325 | 0.0159 |
| LLM-CF | 0.0541 | 0.0860 | 0.0412 |
| LLM-FTAll | 0.0653 | 0.0989 | 0.0520 |
| Llm-FixOrd | 0.0694 | 0.1053 | 0.0524 |
| LLM-PreRec | 0.0639 | 0.1021 | 0.0498 |
| CLLM4Rec | **0.0735** | **0.1149** | **0.0536** |

| Company | Recall@10 | Recall@20 | NDCG@10 |
|---|---|---|---|
| Two-Tower | 0.1186 | 0.2041 | 0.0979 |
| M6-Retrieval | 0.1279 | 0.2118 | 0.1020 |
| CLLM4Rec-Emb | 0.1302 | 0.2165 | 0.1034 |
| CLLM4Rec | **0.1427** | **0.2398** | **0.1199** |

by a large margin. This is because MD-CVAE uses shallow bag-of-words to represent the textual features, for which pretrained LLMs in CLLM4Rec can provide deeper understanding via their pretrained knowledge. The importance of pretrained knowledge can also be shown by the LLM-Scratch model, which performs the worst among all included baselines. An interesting finding is that, LLM-FTAll, which finetunes the whole model including the pretrained LLM backbone, performs worse than CLLM4Rec, which optimizes only the newly introduced user/item token embeddings. The reason could be that, since the weights of the pretrained LLM are fully optimized, the recommendation-specific corpus is still not enough to adapt the pretrained LLM with good generalization ability for RS. Therefore, the cons of degenerating the pretrained knowledge outweigh the introduction of RS-specific knowledge. We can also find that LLM-PreRec, which uses the collaborative LLM in the pretraining stage to generate recommendations,is already a strong baseline. This demonstrates the effectiveness of the soft+hard prompting strategy, which facilitates efficient and stable language modeling on recommendation-oriented corpus with heterogeneous tokens. Still, CLLM4Rec performs better than LLM-PreRec, which shows the effectiveness of recommendation-oriented finetuning in adapting collaborative LLM for efficient recommendations.

*4.2.3* ***Results on the Company Dataset***. In the real-world experiments, we compare CLLM4Rec with the two-tower (TT) model utilized in the Company for job recommendations. The TT model is implemented as a two-branch multi-layer perceptron (MLP), where the input user/item embeddings include embeddings extracted from a graph neural network (GNN) learned on user-job bipartite graph, as well as features extracted from an internal BERT model. In addition, since the textual features are available for almost every user and item, we compare CLLM4Rec with the state-of-the-art LLM-based RS, M6-Retrieval [4], which takes the dimensional-reduced last-layer embeddings of user/item descriptions from M6 Transformer for contrastive recommendations. The results are summarized in Table 2. For Table 2, we can find that CLLM4Rec outperforms the

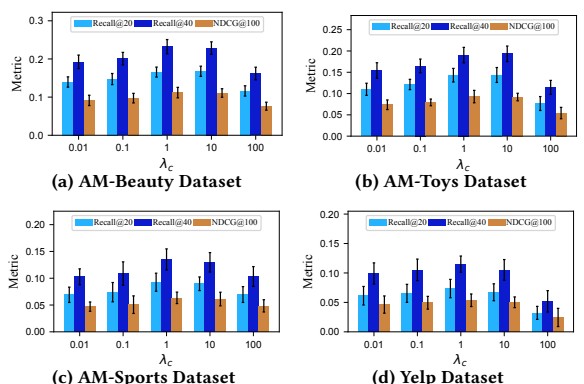

**Figure 4: Sensitivity analysis w.r.t. $\lambda_c$, which controls the strength of mutual-regularization for CLLM4Rec.**

shallow TT model by a large margin. However, although the inference latency for CLLM4Rec is significantly improved compared with existing methods due to the introduction of recommendation-oriented finetuning, directly deploying CLLM4Rec online is still infeasible, as the inference budgets are higher compared to the TT model. Therefore, we design the CLLM4Rec-Emb baseline, which includes the user/item token embeddings $Z^{l,u}$ and $Z^{l,v}$ learned from CLLM4Rec (projected into 128 dimensions) as extra inputs for the TT model, which demonstrates a performance improvement than the original TT model and the M6-Retrieval model in our offline experiment. This demonstrates the potential application of CLLM4Rec in industrial applications where low latency matters.

## 4.3 Parameter Sensitivity Analysis

To further answer **RQs. 2** and **3**, we vary $\lambda_c$ in Eqs. (7), (8), and (10) that controls the strength of mutual regularization and investigates how it influences the performance of CLLM4Rec. From Fig. 4, we can find that, when $\lambda_c$ is small, the mutual regularization is weak, and content LLM cannot provide enough user/item content side information to support the collaborative LLM and RecLLM. Therefore, the recommendation performance degenerates to a similar level as the LLM-CF. On the other hand, when $\lambda_c$ is too large, the MR loss in Eqs. (7), (8) and (10) dominates, which hinders CLLM4Rec from learning user/item token embeddings via language modeling and finetuning. Generally, for all four datasets, the performance of CLLM4Rec peaks at around $\lambda_c = 1$, which serves as a good start when applying the GPT-based CLLM4Rec to new datasets.

## 5 CONCLUSION

In this paper, we proposed CLLM4Rec, the first method that tightly couples the ID paradigm and the LLM paradigm of RS, which faithfully captures user/item semantics while fully utilizing encoded knowledge and logical reasoning ability of pretrained LLMs simultaneously. Specifically, with mutually-regularized pretraining based on soft+hard prompting strategy, CLLM4Rec can effectively capture the user/item collaborative and content information via language modeling. Furthermore, with recommendation-oriented finetuning, the pretrained knowledge of CLLM4Rec can be fully utilized to efficiently generate recommendations. Extensive experiments show the multi-faceted superiority of CLLM4Rec over state-of-the-art.

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

1045
1046
1047
1048
1049
1050
1051
1052
1053
1054
1055
1056
1057
1058
1059
1060
1061
1062
1063
1064
1065
1066
1067
1068
1069
1070
1071
1072
1073
1074
1075
1076
1077
1078
1079
1080
1081
1082
1083
1084
1085
1086
1087
1088
1089
1090
1091
1092
1093
1094
1095
1096
1097
1098
1099
1100
1101
1102
1103
1104
1105
1106
1107
1108
1109
1110
1111
1112
1113
1114
1115
1116
1117
1118
1119
1120
1121
1122
1123
1124
1125
1126
1127
1128
1129
1130
1131
1132
1133
1134
1135
1136
1137
1138
1139
1140
1141
1142
1143
1144
1145
1146
1147
1148
1149
1150
1151
1152
1153
1154
1155
1156
1157
1158
1159
1160

**Table 3: Statistics of the datasets. #Feat. stands for number of textual features (i.e., # reviews for AM/Yelp datasets, and #user biography+#job descriptions for the Company dataset.**

| Dataset | #Int. | #Users | #Items | Sparsity | #Feat. |
|---------|-------|--------|--------|----------|--------|
| AM-Beauty | 94,148 | 10,553 | 6,086 | 99.85% | 70,604 |
| AM-Toys | 95,420 | 11,268 | 7,309 | 99.88% | 70,784 |
| AM-Sports | 185,718 | 22,686 | 12,301 | 99.93% | 137,618 |
| Yelp | 292,017 | 28,330 | 18,775 | 99.94% | 224,825 |
| Company | 90,173 | 22,391 | 1,071 | 99.62% | 23,362 |

**Table 4: Comparison between CLLM4Rec and various baselines with T5-backbone on three Amazon Review datasets.**

| AM-Beauty | Recall@20 | Recall@40 | NDCG@100 |
|-----------|-----------|-----------|----------|
| Multi-VAE | 0.1295 | 0.1720 | 0.0835 |
| MD-CVAE | 0.1472 | 0.2058 | 0.0976 |
| BERT4Rec | 0.1126 | 0.1677 | 0.0781 |
| S³Rec | 0.1354 | 0.1789 | 0.0867 |
| CLLM4Rec-T5 | 0.1538 | 0.2105 | 0.1052 |
| CLLM4Rec | **0.1656** | **0.2323** | **0.1118** |

| AM-Toys | Recall@20 | Recall@40 | NDCG@100 |
|---------|-----------|-----------|----------|
| Multi-VAE | 0.1076 | 0.1558 | 0.0781 |
| MD-CVAE | 0.1291 | 0.1804 | 0.0844 |
| BERT4Rec | 0.0853 | 0.1375 | 0.0532 |
| S³Rec | 0.1064 | 0.1524 | 0.0665 |
| CLLM4Rec-T5 | 0.1328 | 0.1840 | 0.0851 |
| CLLM4Rec | **0.1436** | **0.1933** | **0.0918** |

| AM-Sports | Recall@20 | Recall@40 | NDCG@100 |
|-----------|-----------|-----------|----------|
| Multi-VAE | 0.0659 | 0.0975 | 0.0446 |
| MD-CVAE | 0.0714 | 0.1180 | 0.0514 |
| BERT4Rec | 0.0521 | 0.0701 | 0.0305 |
| S³Rec | 0.0616 | 0.0813 | 0.0438 |
| CLLM4Rec-T5 | 0.0845 | 0.1226 | 0.0589 |
| CLLM4Rec | **0.0926** | **0.1351** | **0.0634** |

## A TECHNICAL DETAILS

### A.1 Implementation of Soft+Hard Prompting

To implement the soft+hard prompting strategy discussed in Section 3.3.2 for decoder-only LLMs such as GPT, we can generate only the "keys" and "values" for the heterogeneous tokens in the prompts $\mathbf{x}_i^{r,p}$, $\mathbf{x}_{ij}^{uv,p}$, and use the "query" of the last token as a start to generate the homogeneous tokens of the main texts $\mathbf{x}_i^{r,m}$, $\mathbf{x}_{ij}^{uv,m}$ for language modeling. For encoder-decoder-based LLMs such as T5, a natural thought is to input the prompts $\mathbf{x}_i^{r,p}$, $\mathbf{x}_{ij}^{uv,p}$ in the encoder, and use the decoder to generate the main texts $\mathbf{x}_i^{r,m}$, $\mathbf{x}_{ij}^{uv,m}$.

### A.2 Recommendation-Oriented Finetuning

If we denote the multinomial probability obtained from the RecLLM prediction head $f_{rec}$ as $\hat{\mathbf{r}}_i^{hold}$, and denote the stacked item

collaborative token embeddings of items interacted by user $i$ as $\mathbf{Z}_i^{l,v}$, the **rec-step** objective of the recommendation-oriented finetuning (regularized with the content LLM) can be formulated as:

$$\mathcal{L}_{\text{rec\_step}}^{\text{MAP}}\left(\mathbf{z}_i^{l,u}, \mathbf{Z}_i^{l,v}, \boldsymbol{\theta}\right) = \underbrace{-\sum_k r_{ik}^{hold} \ln \hat{r}_i^{hold}}_{\text{Multinomial NLL Loss}} \underbrace{- \frac{\lambda_l}{2}\left\|\mathbf{z}_i^{l,u}\right\| - \frac{\lambda_l}{2}\left\|\mathbf{z}_j^{l,v}\right\|}_{\text{Prior loss}}$$

$$\underbrace{- \frac{\lambda_c}{2}\left\|\mathbf{z}_i^{l,u} - \hat{\mathbf{z}}_i^{c,u}\right\|_2^2 - \sum_k \frac{\lambda_c}{2} \cdot \left\|\mathbf{z}_{ik}^{l,v} - \hat{\mathbf{z}}_{ik}^{c,v}\right\|_2^2}_{\text{MR loss with content LLM}} + C_{rec},$$

(10)

where NLL stands for negative log-likelihood, and $C_{rec}$ is the constant irrelevant for the optimization purpose. From the form of the multinomial NLL loss we can find that, when finetuning the RecLLM according to Eq. (10), the $\mathbf{h}_{l,i,-1}^{rec}$ output by the LLM4Rec base model $\hat{llm}_l$, which can be viewed as the user latent variable summarizing the historical interaction of user $i$, is encouraged to be similar to the collaborative embeddings of all the interacted items.

## B EXPERIMENTS

### B.1 Statistics of the Datasets

The statistics of the datasets are summarized in Table 3.

### B.2 Experiments on T5 Backbone

*B.2.1* **Implementation**. We adopt the T5-base model[9] as the backbone, which has 32,128 vocab tokens (the last 28 tokens are empty), where each token is associated with a 768-dimensional vocab embedding. Model training generally follows similar steps as the model with GPT-2 backbone described in Section 4.1.2, where we first warm up the content LLM as Eq. (5) for ten epochs. Then, we conduct the mutually-regularized finetuning as Eqs. (7), (8) for 100 epoch, and conduct finetuning as Eq. (10) for 150 epochs.

*B.2.2* **Results & Analysis**. The experimental results are summarized in Table 4. We can find that although CLLM4Rec with T5 backbone generally outperforms ID-based and shallow LM-based baselines, its performance is consistently worse than CLLM4Rec with GPT-2 backbone. The overall inferior performance of CLLM4Rec with T5 backbone can be two-fold. First, we note that the vocab embeddings in T5 are initialized with unit variance, whereas embeddings in GPT-2 are initialized with a variance of 0.02. Therefore, the weights and embeddings in T5 has much larger numerical values, which leads to large update steps when errors are backpropagating from the outputs to the prompts. Therefore, the training is not as stable as the GPT-2 backbone. In addition, in the finetuning stage of the original T5 model, the prompts are generally used to guide the macro behavior of the model. e.g., changing the model behavior from question answering to machine generation via prompt "translate English to French". Therefore, another reason for the inferiority of T5 backbone could be the mismatch between the original T5 prompts and the prompts intended to be used in CLLM4Rec.

---

[9]https://huggingface.co/t5-base.