# OpenReview forum: "Collaborative Large Language Model for Recommender Systems"
_ACM.org/TheWebConf/2024/Conference — TheWebConf24 Oral_

### Official Review · Reviewer_VPLK · 2023-11-16

**Novelty:** 4
**Technical Quality:** 5

**Review:**

Overview:
This paper introduces CLLM4Rec, a novel generative recommender system that integrates large language models (LLMs) with the
ID paradigm of recommender systems. It addresses the semantic gap in current systems by extending LLM vocabulary with user/
item ID tokens and employing a unique soft+hard prompting strategy during pretraining. This method effectively learns user/item
embeddings through language modeling on specific corpora derived from user-item interactions and features. Additionally, the
paper introduces a mutual regularization strategy and a recommendation-oriented finetuning strategy, significantly enhancing
recommendation effectiveness and efficiency, as demonstrated in experiments on multiple real-world datasets.
Pros:
1) The integration of the ID-based paradigm with Large Language Models (LLMs) represents a significant advancement in research.
2) The introduction section of the paper is well-written.
3) The comprehensive review of LLM and LLM4RS is impressive.
Cons:
Refer to the questions section.

**Questions:**

1. The problem description lacks clarity and is somewhat confusing. It misses essential steps in the
explanation or the necessary formulae. The attention to detail in the diagrams is insufficient.
2. The overall framework appears more as a stack of engineering tricks rather than a cohesive structure
with theoretical backing. There is a notable absence of necessary explanations and theoretical analysis for
the use of each component in the framework.
3. The design of the experimental section is not rational. A significant portion is devoted to comparing
and analyzing performance metrics, while the sensitivity analysis for RQ2 and RQ3 is cursory, spanning
just a few lines. There is a need for more comprehensive validation, such as ablation studies on various
components and interpretability analysis, to convincingly demonstrate the efficacy of your framework.

**Reviewer Confidence:**

3: The reviewer is confident but not certain that the evaluation is correct

**Scope:**

4: The work is relevant to the Web and to the track, and is of broad interest to the community

---

### Official Review · Reviewer_YSDZ · 2023-11-22

**Novelty:** 5
**Technical Quality:** 5

**Review:**

The paper describes the adaptations of Pre-Trained Language models to the domain of recommender systems. There are two key ideas in the paper:
1) Expand Model's vocabulary using additional special tokens for users and items.
2) Add collaborative head to the model, which is specifically trained to predict next interactions in the sequence.

Overall I enjoyed reading the paper, and I found these ideas interesting. I also found that the way material is presented is easy to understand.

There are few issues, which I'd like the authors to address/elaborate though:
1) In my opinion the term Large Language Models is misused in the paper and it is somewhat misleading. Paper uses GPT-2 and T5 as the backbones, and these models the era of LLMs.  A recent well-cited survey on Large Language Models [1] clearly differentiates GPT-2 size models from LLMs: "Although scaling is mainly conducted in model size (with similar architectures and pre-training tasks), these large-sized PLMs display different behaviors from smaller PLMs (e.g., 330M-parameter BERT and 1.5 Bparameter GPT-2) and show surprising abilities (called emergent abilities) in solving a series of complex tasks. For example, GPT-3 can solve few-shot tasks through in-context learning, whereas GPT-2 cannot do well.

2) Data Pollution. GPT-2 was trained using Reddit dataset, and T5 was trained using a version of the Common Crawl dataset. Both of these  datasets contain parts of Amazon Reviews (e. g. for Reddit there is a subreddit with amazon reviews [2]) and Common Crawl also has amazon pages (e. g., project [3]  extracts amazon reviews from common crawl). Given that both train data and test data are constructed using amazon reviews, and the backbone models seen parts of these reviews already, it is hard to say what effect this pollution has in the success of the model.

3) Lack of text-based recommendation baselines. Perhaps P5[4] is the closest competitor of the described approach, and given that the code of a version of P5 is freely available[5], it is strange that the authors choose not to use P5 as a baseline.

[1]Zhao, W.X., Zhou, K., Li, J., Tang, T., Wang, X., Hou, Y., Min, Y., Zhang, B., Zhang, J., Dong, Z. and Du, Y., 2023. A survey of large language models. arXiv preprint arXiv:2303.18223.
[2] https://www.reddit.com/r/amazonreviews/
[3] https://github.com/devvid/python-common-crawl-amazon-example/tree/master
[4] Geng, S., Liu, S., Fu, Z., Ge, Y. and Zhang, Y., 2022, September. Recommendation as language processing (rlp): A unified pretrain, personalized prompt & predict paradigm (p5). In Proceedings of the 16th ACM Conference on Recommender Systems (pp. 299-315).
[5] https://github.com/agiresearch/OpenP5

**Questions:**

Why do you call GPT2 and T5 "large language models"? isn't "Pretrained Language Models" a better term?
How can you make sure that the improvements are not caused by the data pollution ?
Why didn't you use other text-based models, such as P5 as a baseline?

**Reviewer Confidence:**

3: The reviewer is confident but not certain that the evaluation is correct

**Scope:**

4: The work is relevant to the Web and to the track, and is of broad interest to the community

---

### Official Review · Reviewer_pdzC · 2023-11-27

**Novelty:** 4
**Technical Quality:** 2

**Review:**

## Strengths

1. Timely study on large language models and recommender systems.
2. The proposed prompt tuning idea (no need to update model parameters on heterogeneous prompts, but only on homogenous text/item sequences) is interesting and seems promising.
3. The paper is overall well-written and easy-to-follow.
4. Code is available during the reviewing phase.
5. Experiments are conducted on public datasets and real-world company datasets.

## Weaknesses

1. Over-claiming.
    1. Over-claiming on "Large Language Model". The paper is all around large language models and recommender systems, however, the actual model size used in experiments is small. Although the author(s) do not explicitly introduce the model size used in experiments, from line 687 (GPT-2 w/ embedding 768) and the corresponding code, we know that the actual backbone model here is the smallest version of 100+M [gpt2](https://huggingface.co/gpt2). I personally cannot agree that a pre-trained language model (PLM) with 100M parameters can be called a "large language model". Note that 100M gpt2 is even smaller than P5 [7] (223M), which was released in March 2022. The author(s) emphasize how LLMs can do reasoning on prompts while recommending, however, it's questionable whether a 100M PLM has significant reasoning ability.
    2. Over-claiming on model performance. Although the input and output of CLLM4Rec are natural for the sequential recommendation task (i.e., input: user and historical item sequence, output: probabilities of the next item), in Section 4, the models are only evaluated on the direct recommendation task. I'm not saying that PLM-based recommendation models have to be evaluated on the sequential recommendation task, but even limited in the direct recommendation task, the compared baselines can not be called "state-of-the-art". The problems are threefold:
        * (1) only Multi-VAE (and its variant w/ textual features) are included as baselines that were originally designed for direct recommendation. More direct recommendation models, like BPR and EASE, are not considered.
        * (2) sequential recommendation models are compared but are constrained in an experimental setting that these models will perform worse for so-called "fair comparison" reasons.
        * (3) P5, a PLM-based recommendation model for both sequential recommendation and direct recommendation, is not included. The author(s) explain that they do not include P5 because 100 candidates must be included for the direct recommendation task. However, P5 can do sequential recommendation and output the next item probabilities over all the items (via beam search). One can trivially modify P5 (for example, via the same procedure of footnote 8, line 751 - 754) to accommodate it for the direct recommendation task in this paper. I don't think that's a proper reason to not compare CLLM4Rec with P5.
2. I'm kind of confused about why one of pre-training tasks of CLLM4Rec is sequential recommendation, but the model should be fine-tuned to be a direct recommendation model. It could be beneficial if the motivation is explained.
3. It could be beneficial if the author(s) can illustrate which parts of parameters are fixed or tuned in each training stage/task.

**Questions:**

Please refer to "Weaknesses" in "Review".

**Reviewer Confidence:**

4: The reviewer is certain that the evaluation is correct and very familiar with the relevant literature

**Scope:**

4: The work is relevant to the Web and to the track, and is of broad interest to the community

---

### Decision · Program_Chairs · 2024-01-22

**Decision:**

Accept (Oral)

**Comment:**

The authors of this work have attempted to develop a generative RS to integrate LLM and ID paradigms of RS to address the limitations of existing RSs based on LLMs. In the original reviews, one reviewer pointed out that the pre-trained language model in this work might not be called LLM given the small model size. Another reviewer thought the idea is interesting, but agreed that the use of LLM in this paper is a bit misleading. The reviewer additionally pointed out the data pollution issue. Two reviewers suggested comparing the proposed method with more related models. The third reviewer raised concerns about the paper's clarity and the experimental design. During the rebuttal phase, the authors provided additional results and detailed responses to the reviewers' comments. Overall, the work has its merits and might be considered for acceptance if there is room.